# META-LEARNING ACQUISITION FUNCTIONS FOR TRANSFER LEARNING IN BAYESIAN OPTIMIZATION

**Michael Volpp**[1]*
**Lukas P. Fröhlich**[1,2]
**Kirsten Fischer**[1]
**Andreas Doerr**[1,3]
**Stefan Falkner**[1]
**Frank Hutter**[4,1]
**Christian Daniel**[1]

[1]Bosch Center for Artificial Intelligence, Renningen, Germany
[2]ETH Zürich, Zürich, Switzerland
[3]Max Planck Institute for Intelligent Systems, Stuttgart/Tübingen, Germany
[4]University of Freiburg, Germany

## ABSTRACT

Transferring knowledge across tasks to improve data-efficiency is one of the open key challenges in the field of global black-box optimization. Readily available algorithms are typically designed to be universal optimizers and, therefore, often suboptimal for specific tasks. We propose a novel transfer learning method to obtain customized optimizers within the well-established framework of Bayesian optimization, allowing our algorithm to utilize the proven generalization capabilities of Gaussian processes. Using reinforcement learning to meta-train an acquisition function (AF) on a set of related tasks, the proposed method learns to extract implicit structural information and to exploit it for improved data-efficiency. We present experiments on a simulation-to-real transfer task as well as on several synthetic functions and on two hyperparameter search problems. The results show that our algorithm (1) automatically identifies structural properties of objective functions from available source tasks or simulations, (2) performs favourably in settings with both scarse and abundant source data, and (3) falls back to the performance level of general AFs if no particular structure is present.

## 1 INTRODUCTION

Global optimization of black-box functions is highly relevant for a wide range of real-world tasks. Examples include the tuning of hyperparameters in machine learning, the identification of control parameters, or the optimization of system designs. Such applications oftentimes require the optimization of relatively low-dimensional ($\lesssim 10D$) functions where each function evaluation is expensive in either time or cost. Furthermore, there is typically no gradient information available.

In this context of data-efficient global black-box optimization, Bayesian optimization (BO) has emerged as a powerful solution (Močkus, 1975; Brochu et al., 2010; Snoek et al., 2012; Shahriari et al., 2016). BO's data efficiency originates from a probabilistic surrogate model which is used to generalize over information from individual data points. This model is typically given by a Gaussian process (GP), whose well-calibrated uncertainty prediction allows for an informed exploration-exploitation trade-off during optimization. The exact manner of performing this trade-off, however, is left to be encoded in an acquisition function (AF). There is a wide range of AFs available in the literature which are designed to yield universal optimization strategies and therefore come with minimal assumptions about the class of target objective functions.

To achieve optimal data-efficiency on new instances of previously seen tasks, however, it is crucial to incorporate the information obtained from these tasks into the optimization. Therefore, transfer

---

*Correspondence to: `Michael.Volpp@de.bosch.com`

learning is an important and active field of research. Indeed, in many practical applications, optimizations are repeated numerous times in similar settings, underlining the need for specialized optimizers. Examples include hyperparameter optimization which is repeatedly done for the same machine learning model on varying datasets or the optimization of control parameters for a given system with varying physical configurations.

Following recent approaches (Swersky et al., 2013; Feurer et al., 2018; Wistuba et al., 2018), we argue that it is beneficial to perform transfer learning for global black-box optimization in the framework of BO to retain the proven generalization capabilities of its underlying GP surrogate model. To not restrict the expressivity of this model, we propose to implicitly encode the task structure in a specialized AF, i.e., in the optimization strategy. We realize this encoding via a novel method which meta-learns a neural AF, i.e., a neural network representing the AF, on a set of source tasks. The meta-training is performed using reinforcement learning, making the proposed approach applicable to the standard BO setting, where we do not assume access to objective function gradients.

Our contributions are (1) a novel transfer learning method allowing the incorporation of implicit structural knowledge about a class of objective functions into the framework of BO through learned neural AFs to increase data-efficiency on new task instances, (2) an automatic and practical meta-learning procedure for training such neural AFs which is fully compatible with the black-box optimization setting, i.e, not requiring objective function gradients, and (3) the demonstration of the efficiency and practical applicability of our approach on a challenging simulation-to-real control task, on two hyperparameter optimization problems, as well as on a set of synthetic functions.

## 2 RELATED WORK

The general idea of improving the performance or convergence speed of a learning system on a given set of tasks through experience on similar tasks is known as learning to learn, meta-learning or transfer learning and has attracted a large amount of interest in the past while remaining an active field of research (Schmidhuber, 1987; Hochreiter et al., 2001; Thrun and Pratt, 1998; Lake et al., 2016).

In the context of meta-learning optimization, a large body of literature revolves around learning local optimization strategies. One line of work focuses on learning improved optimizers for the training of neural networks, e.g., by directly learning update rules (Bengio et al., 1991; Runarsson and Jonsson, 2000) or by learning controllers for selecting appropriate step sizes for gradient descent (Daniel et al., 2016). Another direction of research considers the more general setting of replacing the gradient descent update step by neural networks which are trained using either reinforcement learning (Li and Malik, 2016; 2017) or in a supervised fashion (Andrychowicz et al., 2016; Metz et al., 2019). Finn et al. (2017), Nichol et al. (2018), and Flennerhag et al. (2019) propose approaches for initializing machine learning models through meta-learning to be able to solve new learning tasks with few gradient steps.

We are currently aware of only one work tackling the problem of meta-learning global black-box optimization (Chen et al., 2017). In contrast to our proposed method, the authors assume access to gradient information and choose a supervised learning approach, representing the optimizer as a recurrent neural network operating on the raw input vectors. Based on statistics of the optimization history accumulated in its memory state, this network directly outputs the next query point. In contrast, we consider transfer learning applications where gradients are typically not available.

A number of articles address the problem of increasing BO's data-efficiency via transfer learning, i.e., by incorporating information obtained from similar optimizations on source tasks into the current target task. A range of methods accumulate all available source and target data in a single GP and make the data comparable via a ranking algorithm (Bardenet et al., 2013), standardization or multi-kernel GPs (Yogatama and Mann, 2014), multi-task GPs (Swersky et al., 2013), the GP noise model (Theckel Joy et al., 2016), or by regressing on prediction biases (Shilton et al., 2017). These approaches naturally suffer from the cubic scaling behaviour of GPs, which can be tackled for instance by replacing the GP model, e.g., with Bayesian neural networks with task-specific embedding vectors (Springenberg et al., 2016) or with adaptive Bayesian linear regression with basis functions shared across tasks via a neural network (Perrone et al., 2018). Recently, Garnelo et al. (2018) proposed Neural Processes as another interesting alternative for GPs with improved scaling behavior. Other

approaches retain the GP surrogate model and combine individual GPs for source and target tasks in an ensemble model with the weights adjusted according to the GP uncertainties (Schilling et al., 2016), dataset similarities (Wistuba et al., 2016), or estimates of the GP generalization performance on the target task (Feurer et al., 2018). Similarly, Golovin et al. (2017) form a stack of GPs by iteratively regressing onto the residuals w.r.t. the most recent source task. In contrast to our proposed method, many of these approaches rely on hand-engineered dataset features to measure the relevance of source data for the target task. Such features have also been used to pick promising initial configurations for BO (Feurer et al., 2015a;b).

The method being closest in spirit and capability to our approach is proposed by Wistuba et al. (2018). It is similar to the aforementioned ensemble techniques with the important difference that the source and target GPs are not combined via a surrogate model but via a new AF, the so-called transfer acquisition function (TAF). This AF is defined to be a weighted superposition of the predicted improvements according to the source GPs and the expected improvement according to the target GP. Viewed in this context, our method also combines knowledge from source and target tasks in a new AF which we represent as a neural network. Our weighting of source and target data is implicitly determined in a meta-learning phase and is automatically regulated during the optimization on the target task to adapt online to the specific objective function at hand. Furthermore, our method does not store and evaluate many source GPs because the knowledge from the source datasets is encoded directly in the network weights of the learned AF. This allows our method to incorporate large amounts of source data while the applicability of TAF is restricted to a comparably small number of source tasks.

## 3 PRELIMINARIES

We are aiming to find a global optimum $\boldsymbol{x}^* \in \arg\max_{\boldsymbol{x} \in \mathcal{D}} f(\boldsymbol{x})$ of some unknown objective function $f : \mathcal{D} \to \mathbb{R}$ on the domain $\mathcal{D} \subset \mathbb{R}^D$. The only means of acquiring information about $f$ is via (possibly noisy) evaluations at points in $\mathcal{D}$. Therefore, at each optimization step $t \in \{1, 2, \dots\}$, the optimizer has to decide for the iterate $\boldsymbol{x}_t \in \mathcal{D}$ solely based on the *optimization history* $\mathcal{H}_t \equiv \{\boldsymbol{x}_i, y_i\}_{i=1}^{t-1}$ with $y_i = f(\boldsymbol{x}_i) + \epsilon$. Here, $\epsilon \sim \mathcal{N}\left(0, \sigma_n^2\right)$ denotes independent and identically distributed Gaussian noise. In particular, the optimizer does not have access to gradients of $f$. To assess the performance of global optimization algorithms, it is natural to use the *simple regret* $R_t \equiv f(\boldsymbol{x}^*) - f(\boldsymbol{x}_t^+)$ where $\boldsymbol{x}_t^+$ is the input location corresponding to the best evaluation found by an algorithm up to and including step $t$. The proposed method relies on the framework of BO and is trained using reinforcement learning. Therefore, we now shortly introduce these frameworks.

**Bayesian Optimization** In Bayesian optimization (BO) (Shahriari et al., 2016), one specifies a prior belief about the objective function $f$ and at each step $t$ builds a probabilistic surrogate model conditioned on the current optimization history $\mathcal{H}_t$. Typically, a Gaussian process (GP) (Rasmussen and Williams, 2005) is employed as the surrogate model in which case the resulting posterior belief about $f(\boldsymbol{x})$ follows a Gaussian distribution with mean $\mu_t(\boldsymbol{x}) \equiv \mathbb{E}\{f(\boldsymbol{x}) \mid \mathcal{H}_t\}$ and variance $\sigma_t^2(\boldsymbol{x}) \equiv \mathbb{V}\{f(\boldsymbol{x}) \mid \mathcal{H}_t\}$, for which closed-form expressions are available. To determine the next iterate $\boldsymbol{x}_t$ based on the belief about $f$ given $\mathcal{H}_t$, a sampling strategy is defined in terms of an *acquisition function* (AF) $\alpha_t(\cdot \mid \mathcal{H}_t) : \mathcal{D} \to \mathbb{R}$. The AF outputs a score value at each point in $\mathcal{D}$ such that the next iterate is defined to be given by $\boldsymbol{x}_t \in \arg\max_{\boldsymbol{x} \in \mathcal{D}} \alpha_t(\boldsymbol{x} \mid \mathcal{H}_t)$. The strength of the resulting optimizer is largely based upon carefully designing the AF to trade-off exploration of unknown versus exploitation of promising areas in $\mathcal{D}$.

There is a wide range of general-purpose AFs available in the literature. Popular choices are *probability of improvement* (PI) (Kushner, 1964), *GP-upper confidence bound* (GP-UCB) (Srinivas et al., 2010), and *expected improvement* (EI) (Močkus, 1975). In our experiments, we will use EI as a not pre-informed baseline AF, so we state its definition here,

$$\mathrm{EI}_t(\boldsymbol{x}) \equiv \mathbb{E}_{f(\boldsymbol{x})}\big\{\max\big[f(\boldsymbol{x}) - f(\boldsymbol{x}_{t-1}^+), 0\big] \,\big|\, \mathcal{H}_t\big\}, \tag{1}$$

and note that it can be written in closed form if $f(\boldsymbol{x})$ follows a Gaussian distribution.

To perform transfer learning in the context of BO, Wistuba et al. (2018) introduced the *transfer acquisition framework* (TAF) which defines a new AF as a weighted superposition of EI on the target

task and the predicted improvements on the source tasks, i.e.,

$$\text{TAF}_t(\boldsymbol{x}) \equiv \frac{w_{M+1}\text{EI}_t^{M+1}(\boldsymbol{x}) + \sum_{j=1}^{M} w_j I_t^j(\boldsymbol{x})}{\sum_{j=1}^{M+1} w_j}, \tag{2}$$

with the predicted improvement

$$I_t^j(\boldsymbol{x}) \equiv \max\left(\mu^j(\boldsymbol{x}) - y_{t-1}^{j,\max}, 0\right). \tag{3}$$

TAF stores separate GP surrogate models for the source and target tasks, with $j \in \{1, \ldots, M\}$ indexing the source tasks and $j = M + 1$ indexing the target task. Therefore, $\text{EI}_t^{M+1}$ denotes EI according to the target GP surrogate model and $\mu^j$ denotes the mean function of the $j$-th source GP model. $y_t^{j,\max}$ denotes the maximum of the mean predictions of the $j$-th source GP model on the set of iterates $\{\boldsymbol{x}_i\}_{i=1}^t$. The weights $w_j \in \mathbb{R}$ are determined either based on the predicted variances of the source and target GP surrogate models (TAF-ME) or, alternatively, by a pairwise comparison of the predicted performance ranks of the iterates (TAF-R).

**Reinforcement Learning**   Reinforcement learning (RL) allows an agent to learn goal-oriented behavior via trial-and-error interactions with its environment (Sutton and Barto, 1998). This interaction process is formalized as a Markov decision process: at step $t$ the agent senses the environment's state $s_t \in \mathcal{S}$ and uses a policy $\pi : \mathcal{S} \to \mathcal{P}(\mathcal{A})$ to determine the next action $a_t \in \mathcal{A}$. Typically, the agent explores the environment by means of a probabilistic policy, i.e., $\mathcal{P}(\mathcal{A})$ denotes the probability measures over $\mathcal{A}$. The environment's response to $a_t$ is the next state $s_{t+1}$, which is drawn from a probability distribution with density $p(s_{t+1} \,|\, s_t, a_t)$. The agent's goal is formulated in terms of a scalar reward $r_t = r(s_t, a_t, s_{t+1})$, which the agent receives together with $s_{t+1}$. The agent aims to maximize the expected cumulative discounted future reward $\eta(\pi)$ when acting according to $\pi$ and starting from some state $s_0 \in \mathcal{S}$, i.e., $\eta(\pi) \equiv \mathbb{E}_\pi\left[\sum_{t=1}^{T} \gamma^{t-1} r_t \,\middle|\, s_0\right]$. Here, $T$ denotes the episode length and $\gamma \in (0, 1]$ is a discount factor.

## 4   METABO ALGORITHM

We devise a global black-box optimization method that is able to automatically identify and exploit structural properties of a given class of objective functions for improved data-efficiency. We stay within the framework of BO, enabling us to exploit the powerful generalization capabilities of a GP surrogate model. The actual optimization strategy which is informed by this GP is classically encoded in a hand-designed AF. Instead, we meta-train on a set of source tasks to replace this AF by a neural network but retain all other elements of the proven BO-loop (middle panel of Fig. 1). To distinguish the learned AF from a classical AF $\alpha_t$, we call such a network a *neural acquisition function* and denote it by $\alpha_{t,\theta}$, indicating that it is parametrized by a vector $\theta$. We dub the resulting algorithm *MetaBO*.

Let $\mathcal{F}$ be the class of objective functions for which we aim to learn a neural acquisition function $\alpha_{t,\theta}$. For instance, $\mathcal{F}$ may be the set of objective functions resulting from different physical configurations of a laboratory experiment or from evaluating the loss function of a machine learning model on different data sets. Often, such objective functions share structure which we aim to exploit for data-efficient optimization on further instances from the same function class. In many relevant cases, it is straightforward to obtain approximations to $\mathcal{F}$, i.e., a set of functions $\mathcal{F}'$ which capture relevant properties of $\mathcal{F}$ but are much cheaper to evaluate (e.g., by using numerical simulations or results from previous hyperparameter optimization tasks (Wistuba et al., 2018)). During an offline meta-training phase, MetaBO makes use of such cheap approximations to identify the implicit structure of $\mathcal{F}$ and to adapt $\theta$ to obtain a data-efficient optimization strategy customized to $\mathcal{F}$.

Typically, the minimal set of inputs to AFs in BO is given by the pointwise GP posterior prediction $\mu_t(\boldsymbol{x})$ and $\sigma_t(\boldsymbol{x})$. To perform transfer learning, the AF has to be able to identify relevant structure shared by the objective functions in $\mathcal{F}$. In our setting, this is achieved via extending this basic set of inputs by additional features which enable the neural AF to evaluate sample locations. Therefore, in addition to the mean $\mu_t(\boldsymbol{x})$ and variance $\sigma_t(\boldsymbol{x})$ at potential sample locations, the neural AF also receives the input location $\boldsymbol{x}$ itself. Furthermore, we add to the set of input features the current

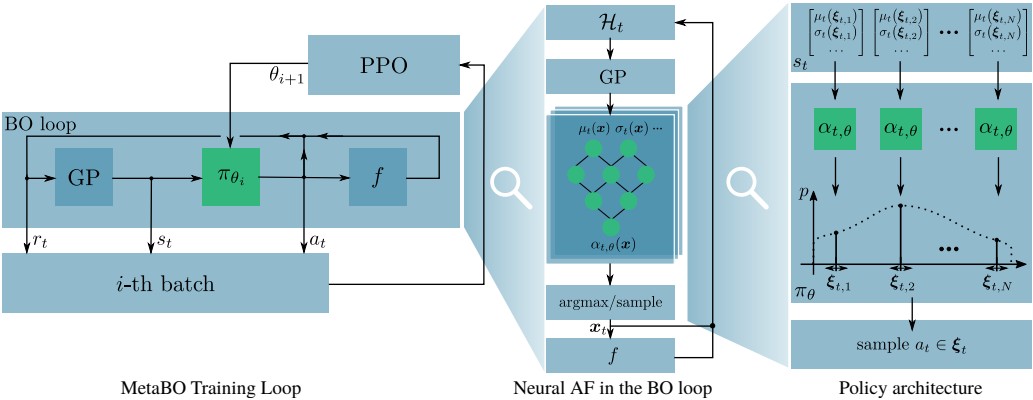

Figure 1: Different levels of the MetaBO framework. Left panel: structure of the training loop for meta-learning neural AFs using RL (PPO). Middle panel: the classical BO loop with a neural AF $\alpha_{t,\theta}$. At test time, there is no difference to classical BO, i.e., $\boldsymbol{x}_t$ is given by the $\arg\max$ of the AF output. During training, the AF corresponds to the RL policy evaluated on an adaptive set $\boldsymbol{\xi}_t \subset \mathcal{D}$. The outputs are interpreted as logits of a categorical distribution from which the actions $a_t = \boldsymbol{x}_t \in \boldsymbol{\xi}_t$ are sampled. This sampling procedure is detailed in the right panel. We indicate by the dotted curve and tiny two-headed arrows that $\alpha_{t,\theta}$ is a function defined on the whole domain $\mathcal{D}$ which can be evaluated at arbitrary points $\boldsymbol{\xi}_{t,n}$ to form the categorical distribution representing the policy $\pi_\theta$.

Table 1: The MetaBO setting in the RL framework.

| RL | MetaBO |
|---|---|
| Policy $\pi_\theta$ | Neural AF $\alpha_{t,\theta}$ |
| Episode | Optimization run on $f \in \mathcal{F}'$ |
| Episode length $T$ | Optimization budget $T$ |
| State $s_t$ | $\left[\mu_t(\boldsymbol{\xi}_{t,n}), \sigma_t(\boldsymbol{\xi}_{t,n}), \boldsymbol{\xi}_{t,n}, t, T\right]_{n=1}^{N}$ |
| Action $a_t$ | Sampling point $\boldsymbol{x}_t \in \boldsymbol{\xi}_t$ |
| Reward $r_t$ | Negative simple regret $-R_t$ |
| Transition $p(s_{t+1} \mid s_t, a_t)$ | Noisy evaluation of $f$, GP update |

optimization step $t$ and the optimization budget $T$, as these features can be valuable for adjusting the exploration-exploitation trade-off (Srinivas et al., 2010). Therefore, we define

$$\alpha_{t,\theta}(\boldsymbol{x}) \equiv \alpha_{t,\theta}[\mu_t(\boldsymbol{x}), \sigma_t(\boldsymbol{x}), \boldsymbol{x}, t, T]. \tag{4}$$

This architecture allows learning a scalable neural AF, as we still base our architecture only on the pointwise GP posterior prediction. Furthermore, neural AFs of this form can be used as a plug-in feature in any state-of-the-art BO framework. In particular, if differentiable activation functions are chosen, a neural AF constitutes a differentiable mapping $\mathcal{D} \rightarrow \mathbb{R}$ and standard gradient-based optimization strategies can be used to find its maximum in the BO loop during evaluation. We further emphasize that after the training phase the resulting neural AF is fully defined, i.e., there is no need to calibrate any AF-related hyperparameters.

**Training Procedure** In the general BO setting, gradients of $\mathcal{F}$ are assumed to be unavailable. This is oftentimes also true for the functions in $\mathcal{F}'$, for instance, when $\mathcal{F}'$ comprises numerical simulations or results from previous optimization runs. Therefore, we resort to RL as the meta-algorithm, as it does not require gradients of the objective functions. Specifically, we use the Proximal Policy Optimization (PPO) algorithm as proposed in Schulman et al. (2017). Tab. 1 translates the MetaBO-setting into RL parlance.

We aim to shape the mapping $\alpha_{t,\theta}(\boldsymbol{x})$ during meta-training in such a way that its maximum location corresponds to a promising sampling location $\boldsymbol{x}$ for optimization. The meta-algorithm PPO explores its state space using a parametrized stochastic policy $\pi_\theta$ from which the actions $a_t = \boldsymbol{x}_t$ are sampled depending on the current state $s_t$, i.e., $a_t \sim \pi_\theta(\cdot \mid s_t)$. As the meta-algorithm requires access to

the global information contained in the GP posterior prediction, the state $s_t$ at optimization step $t$ *formally* corresponds to the *functions* $\mu_t$ and $\sigma_t$ (together with the aforementioned additional input features to the neural AF). To connect the neural AF $\alpha_{t,\theta}$ with the policy $\pi_\theta$ and to arrive at a practical implementation, we evaluate $\mu_t$ and $\sigma_t$ on a discrete set of points $\boldsymbol{\xi}_t \equiv \{\boldsymbol{\xi}_{t,n}\}_{n=1}^N \subset \mathcal{D}$ and feed these evaluations through the neural AF $\alpha_{t,\theta}$ one at a time, yielding one scalar output value $\alpha_{t,\theta}(\boldsymbol{\xi}_{t,n}) = \alpha_{t,\theta}[\mu_t(\boldsymbol{\xi}_{t,n}), \sigma_t(\boldsymbol{\xi}_{t,n}), \boldsymbol{\xi}_{t,n}, t, T]$ for each point $\boldsymbol{\xi}_{t,n}$. These outputs are interpreted as the logits of a categorical distribution, i.e., we arrive at the policy architecture

$$\pi_\theta\left(\cdot \mid s_t\right) \equiv \mathrm{Cat}\left[\alpha_{t,\theta}(\boldsymbol{\xi}_{t,1}), \ldots, \alpha_{t,\theta}(\boldsymbol{\xi}_{t,N})\right], \tag{5}$$

cf. Fig. 1, right panel. Therefore, the proposed policy evaluates the *same* neural acquisition function $\alpha_{t,\theta}$ at arbitrarily many input locations $\boldsymbol{\xi}_{t,n}$ and preferably samples actions $\boldsymbol{x}_t \in \boldsymbol{\xi}_t$ with high $\alpha_{t,\theta}(\boldsymbol{x}_t)$. This incentivizes the meta-algorithm to adjust $\theta$ such that promising locations $\boldsymbol{\xi}_{t,n}$ are attributed high values of $\alpha_{t,\theta}(\boldsymbol{\xi}_{t,n})$.

Calculating a sufficiently fine *static* set $\boldsymbol{\xi}$ of of evaluation points is challenging for higher dimensional settings. Instead, we build on the approach proposed by Snoek et al. (2012) and continuously adapt $\boldsymbol{\xi} = \boldsymbol{\xi}_t$ to the current state of $\alpha_{t,\theta}$. At each step $t$, $\alpha_{t,\theta}$ is first evaluated on a static and relatively coarse Sobol grid (Sobol, 1967) $\boldsymbol{\xi}_{\mathrm{global}}$ spanning the whole domain $\mathcal{D}$. Subsequently, local maximizations of $\alpha_{t,\theta}$ are started from the $k$ points corresponding to the best evaluations. We denote the resulting set of local maxima by $\boldsymbol{\xi}_{\mathrm{local},t}$. Finally, we define $\boldsymbol{\xi}_t \equiv \boldsymbol{\xi}_{\mathrm{local},t} \cup \boldsymbol{\xi}_{\mathrm{global}}$. The adaptive local part of this set enables the RL agent to exploit what it has learned so far by picking points which look promising according to the current neural AF while the static global part maintains exploration. We refer the reader to App. B.1 for details.

The final characteristics of the neural AF are controlled through the choice of reward function. For the presented experiments we emphasized fast convergence to the optimum by using the negative simple regret as the reward signal, i.e., we set $r_t \equiv -R_t$.[1] This choice does not penalize explorative evaluations which do not yield an immediate improvement and additionally serves as a normalization of the functions $f \in \mathcal{F}'$. We emphasize that the knowledge of the true maximum is only required during training and that cases in which it is not known at training time do not limit the applicability of our method, as a cheap approximation (e.g., by evaluating the function on a coarse grid) can also be utilized.

The left panel of Fig. 1 depicts the resulting training loop graphically. The outer loop corresponds to the RL meta-training iterations, each performing a policy update step $\pi_{\theta_i} \to \pi_{\theta_{i+1}}$. To approximate the gradients of the PPO loss function, we record a batch of episodes in the inner loop, i.e., a set of $(s_t, a_t, r_t)$-tuples, by rolling out the current policy $\pi_{\theta_i}$. At the beginning of each episode, we draw some function $f$ from the training set $\mathcal{F}'$ and fix an optimization budget $T$. In each iteration of the inner loop we determine the adaptive set $\boldsymbol{\xi}_t$ and feed the state $s_t$ through the policy which yields the action $a_t = \boldsymbol{x}_t$. We then evaluate $f$ at $\boldsymbol{x}_t$ and use the result to compute the reward $r_t$ and to update the optimization history: $\mathcal{H}_t \to \mathcal{H}_{t+1} = \mathcal{H}_t \cup \{\boldsymbol{x}_t, y_t\}$. Finally, the GP is conditioned on the updated optimization history $\mathcal{H}_{t+1}$ to obtain the next state $s_{t+1}$.

## 5 EXPERIMENTS

We trained MetaBO on a wide range of function classes and compared the performance of the resulting neural AFs with the general-purpose AF expected improvement (EI)[2] as well as the transfer acquisition function framework (TAF) which proved to be the current state-of-the-art solution for transfer learning in BO in an extensive experimental study (Wistuba et al., 2018). We tested both the ranking-based version (TAF-R) and the mixture-of-experts version (TAF-ME). We refer the reader to App. A for a more detailed experimental investigation of MetaBO's performance.

If not stated differently, we report performance in terms of the median simple regret $R_t$ over 100 optimization runs on unseen test functions as a function of the optimization step $t$ together with $30\%/70\%$ percentiles (shaded areas). We emphasize that all experiments use the same MetaBO

---

[1]Alternatively, a logarithmically-transformed version of this reward signal, $r_t \equiv -\log_{10} R_t$, can be used in situations where high-accuracy solutions shall be rewarded.

[2]We also evaluated probability of improvement (PI) as well as GP-upper confidence bound (GP-UCB) but do not present the results here to avoid clutter, as EI performed better in all our experiments.

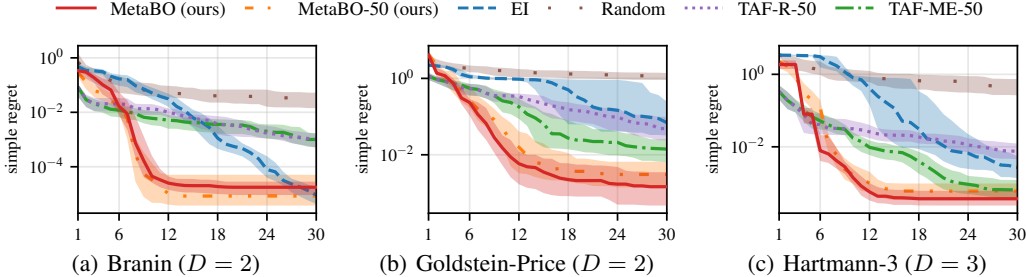

Figure 2: Performance on three global optimization benchmark functions with random translations sampled uniformly from $[-0.1, 0.1]^D$ and scalings from $[0.9, 1.1]$. To test TAF's performance, we randomly picked $M = 50$ source tasks from this function class and evaluated both the ranking-based version (TAF-R-50) and the mixture-of-experts version (TAF-ME-50). We trained MetaBO on the same set of source tasks (MetaBO-50). In contrast to TAF, MetaBO can also be trained without manually restricting the set of available source tasks. The corresponding results are labelled "MetaBO". MetaBO outperformed EI by clear margin, especially in early stages of the optimization. After few steps used to identify the specific instance of the objective function, MetaBO also outperformed both flavors of TAF over wide ranges of the optimization budget. Results for TAF-20 can be found in App. A.4, Fig. 12.

hyperparameters, making our method easily applicable in practice. Furthermore, MetaBO does not increase evaluation time considerably compared to standard AFs, cf. App. A.2, Tab. 3. In addition, even the most expensive of our experiments (the simulation-to-real task, due to the simulation in the BO loop) required not more than 10h of training time on a moderately complex architecture (10 CPU workers, 1 GPU), which is fully justified for our intended offline transfer learning use-case. To foster reproducibility, we provide a detailed exposition of the experimental settings in App. B and make the source code of MetaBO available online.[3]

**Global Optimization Benchmark Functions**    We evaluated our method on a set of synthetic function classes based on the standard global optimization benchmark functions Branin ($D = 2$), Goldstein-Price ($D = 2$), and Hartmann-3 ($D = 3$) (Picheny et al., 2013). To construct the training set $\mathcal{F}'$, we applied translations in $[-0.1, 0.1]^D$ as well as scalings in $[0.9, 1.1]$.

As TAF stores and evaluates one source GP for each source task, its applicability is restricted to a relatively small amount of source data. For the evaluations of TAF and MetaBO, we therefore picked a random set of $M = 50$ source tasks from the continuously parametrized family $\mathcal{F}'$ of available objective functions and spread these tasks uniformly over the whole range of translations and scalings (MetaBO-50, TAF-R-50, TAF-ME-50). We used $N_{\text{TAF}} = 100$ data points for each source GP of TAF. We also tested both flavors of TAF for $M = 20$ source tasks (with $N_{\text{TAF}} = 50$) and observed that TAF's performance does not necessarily increase with more source data, rendering the choice of suitable source tasks cumbersome. Fig. 2 shows the performance on unseen functions drawn randomly from $\mathcal{F}'$. To avoid clutter, we move the results for TAF-20 to App. A.4, cf. Fig. 12. MetaBO-50 outperformed EI by large margin, in particular at early stages of the optimization, by making use of the structural knowledge about $\mathcal{F}'$ acquired during the meta-learning phase. Furthermore, MetaBO-50 outperformed both flavors of TAF-50 over wide ranges of the optimization budget. This is due to its ability to learn sampling strategies which go beyond a combination of a prior over $\mathcal{D}$ and a standard AF (as is the case for TAF). Indeed, note that MetaBO spends some initial non-greedy evaluations to identify specific properties of the target objective function, resulting in much more efficient optimization strategies. We investigate this behaviour further on simple toy experiments and using easily interpretable baseline AFs in App. A.1.

We further emphasize that MetaBO does not require the user to manually pick a suitable set of source tasks but that it can naturally learn from the whole set $\mathcal{F}'$ of available source tasks by randomly picking a new task from $\mathcal{F}'$ at the beginning of each BO iteration and aggregating this information in the neural AF weights. We also trained this full version of MetaBO (labelled "MetaBO") on the

---

[3]https://github.com/boschresearch/MetaBO

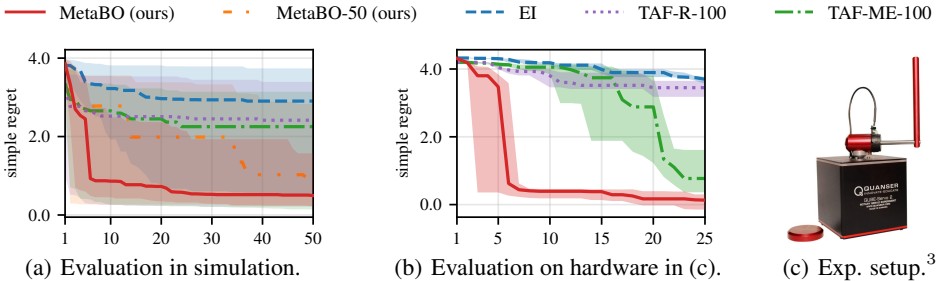

(a) Evaluation in simulation.  (b) Evaluation on hardware in (c).  (c) Exp. setup.[3]

Figure 3: Performance on a simulation-to-real task (cf. text). MetaBO and TAF used source data from a cheap numerical simulation. (a) Performance on an extended training set in simulation. (b) Transfer to the hardware depicted in (c), averaged over ten BO runs. MetaBO learned robust neural AFs with very strong optimization performance and online adaption to the target objectives, which reliably yielded stabilizing controllers after less than ten BO iterations while TAF-ME-100, TAF-R-100, and EI explore too heavily. Comparing the results for MetaBO and MetaBO-50 in simulation, we observe that MetaBO benefits from its ability to learn from the whole set of available source data, while TAF's applicability is restricted to a comparably small number of source tasks. We move the results for TAF-50 to App. A.4, Fig. 13.

global optimization benchmark functions, obtaining performance comparable with MetaBO-50. We demonstrate below that for more complex experiments, such as the simulation-to-real task, MetaBO's ability to learn from the full set of available source tasks is crucial for efficient transfer learning. We also investigate the dependence of MetaBO's performance on the number of source tasks in more detail in App. A.2.

As a final test on synthetic functions, we evaluated the neural AFs on objective functions outside of the training distribution. This can give interesting insights into the nature of the problems under consideration. We move the results of this experiment to App. A.3.

**Simulation-to-Real Task** Sample efficiency is of special interest for the optimization of real world systems. In cases where an approximate model of the system can be simulated, the proposed approach can be used to improve the data-efficiency on the real system. To demonstrate this, we evaluated MetaBO on a $4D$ simulation-to-real experiment. The task was to stabilize a Furuta pendulum (Furuta et al., 1992) for $5\,\mathrm{s}$ around the upper equilibrium position using a linear state-feedback controller. We applied BO to tune the four feedback gains of this controller (Fröhlich et al., 2020). To assess the performance of a given controller, we employed a logarithmic quadratic cost function (Bansal et al., 2017). If the controller was not able to stabilize the system or if the voltage applied to the motor exceeded some safety limit, we added a penalty term proportional to the remaining time the pendulum would have had to be stabilized for successfully completing the task. We emphasize that the cost function is rather sensitive to the control gains, resulting in a challenging black-box optimization problem.

To meta-learn the neural AF, we employed a fast numerical simulation based on the nonlinear dynamics equations of the Furuta pendulum which only contained the most basic physical effects. In particular, effects like friction and stiction were not modeled. The training distribution was generated by sampling the physical parameters of this simulation (two lengths, two masses), uniformly on a range of $75\% - 125\%$ around the measured parameters of the hardware (Quanser QUBE – Servo 2,[4] Fig. 3(c)). We also used this simulation to generate $M = 100$ source tasks for TAF ($N_{\mathrm{TAF}} = 200$).

Fig. 3(a) shows the performance on objective functions from simulation. Again, MetaBO learned a sophisticated sampling strategy which first identifies the target objective function and adapts its optimization strategy accordingly, resulting in very strong optimization performance. In contrast, TAF's superposition of a prior over $\mathcal{D}$ obtained from the source tasks with EI on the target task leads to excessive explorative behaviour. We move further experimental results for TAF-50 to App. A.4, Fig. 13.

---

[4]https://www.quanser.com/products/qube-servo-2

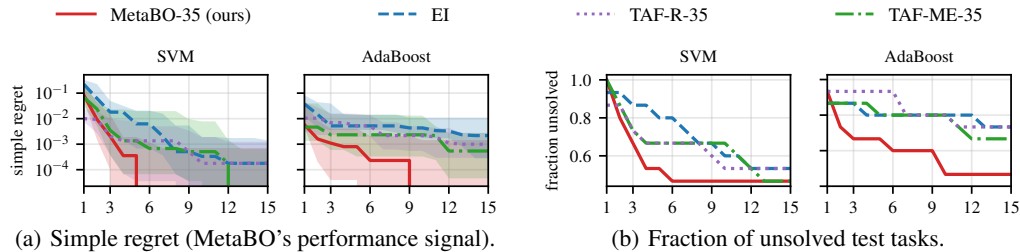

Figure 4: Performance on two $2D$ hyperparameter optimization tasks (SVM and AdaBoost). We trained MetaBO on precomputed data for 35 randomly chosen datasets and used the same datasets as source tasks for TAF. The remaining 15 datasets were used for this evaluation. MetaBO learned very data-efficient sampling strategies on both experiments, outperforming the benchmark methods by clear margin. Note that the optimization domain is discrete and therefore tasks can be solved exactly, corresponding to zero regret.

By comparing the performance of MetaBO and MetaBO-50 in simulation, we find that our architecture's ability to incorporate large amounts of source data is indeed beneficial on this complex optimization problem. The results in App. A.2 underline that this task indeed requires large amounts of source data to be solved efficiently. This is substantiated by the results on the hardware, on which we evaluated the full version of MetaBO and the baseline AFs obtained by training on data from simulation without any changes. Fig. 3(b) shows that MetaBO learned a neural AF which generalizes well from the simulated objectives to the hardware task and was thereby able to rapidly adjust to its specific properties. This resulted in very data-efficient optimization on the target system, consistently yielding stabilizing controllers after less than ten BO iterations. In comparison, the benchmark AFs required many samples to identify promising regions of the search space and therefore did not reliably find stabilizing controllers within the budget of 25 optimization steps.

As it provides interesting insights into the nature of the studied problem, we investigate MetaBO's generalization performance to functions outside of the training distribution in App. A.3. We emphasize, however, that the intended use case of our method is on unseen functions drawn from the training distribution. Indeed, by measuring the physical parameters of the hardware system and adjusting the ranges from which the parameters are drawn to generate $\mathcal{F}'$ according to the measurement uncertainty, the training distribution can be modelled in such a way that the true system parameters lie inside of it with high confidence.

**Hyperparameter Optimization** We tested MetaBO on two $2D$-hyperparameter optimization (HPO) problems for RBF-based SVMs and AdaBoost. As proposed in Wistuba et al. (2018), we used precomputed results of training these models on 50 datasets[5] with 144 parameter configurations (RBF kernel parameter, penalty parameter $C$) for the SVMs and 108 configurations (number of product terms, number of iterations) for AdaBoost. We randomly split these datasets into 35 source datasets used for training MetaBO as well as for TAF and evaluated the resulting optimization strategies on the remaining 15 datasets. To determine when to stop the meta-training of MetaBO, we performed 7-fold cross validation on the training datasets. We emphasize that MetaBO did not use more source data than TAF in this experiment, underlining again its broad applicability in situations with both scarse and abundant source data. The results (Fig. 4) show that MetaBO learned very data-efficient neural AFs which surpassed EI und TAF on both experiments.

**General Function Classes** Finally, we evaluated the performance of MetaBO on function classes without any particular structure except a bounded correlation lengthscale. As there is only little structure present in this function class which could be exploited in the transfer learning setting, it is desirable to obtain neural AFs which fall back at least on the performance level of general-purpose AFs such as EI. We performed two different experiments of this type. For the first experiment, we sampled the objective functions from a GP prior with squared-exponential (RBF) kernel with lengthscales drawn uniformly from $\ell \in [0.05, 0.5]$.[6] For the second experiment, we used a GP prior

---

[5]Visualizations of the objective functions can be found on `http://www.hylap.org`

[6]We normalized the optimization domain to $\mathcal{D} = [0, 1]^D$.

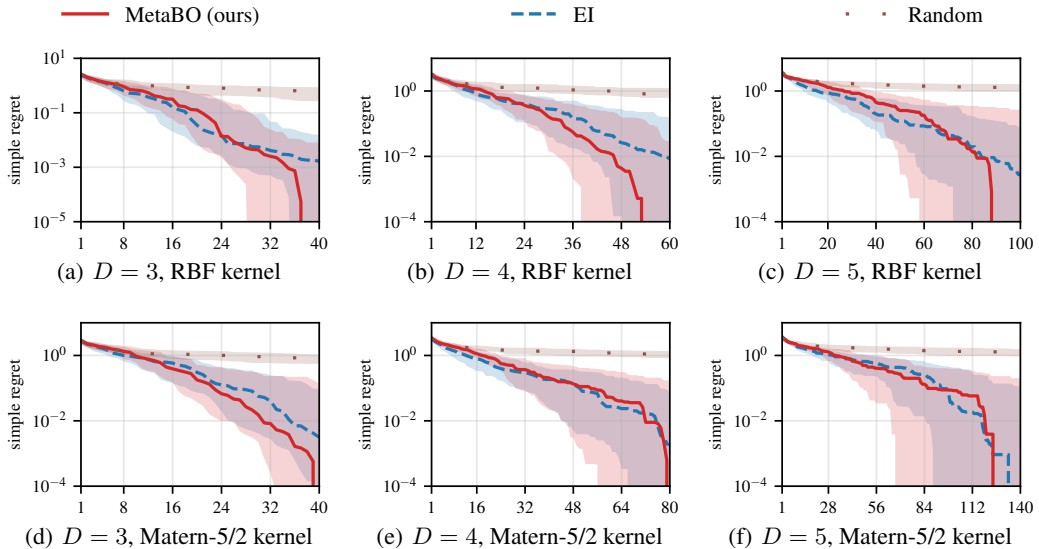

Figure 5: Performance of MetaBO trained on $D = 3$-dimensional objective functions sampled from a GP prior with RBF kernel (upper row) and Matern-5/2 kernel (lower row) with lengthscales drawn randomly from $\ell \in [0.05, 0.5]$. Panels (a, d) show the performance on these training distributions. As we excluded the $x$-feature from the neural AF inputs during training, the resulting AFs can be applied to functions of different dimensionalities. We evaluated each AF on $D = 4$ and $D = 5$ without retraining MetaBO. We report simple regret w.r.t. the best observed function value, determined separately for each function in the test set.

with Matern-5/2 kernel with the same range of lengthscales. For the latter experiment we also used the Matern-5/2 kernel (in contrast to the RBF kernel used in all other experiments) as the kernel of the GP surrogate model to avoid model mismatch. For both types of function classes we trained MetaBO on $D = 3$ dimensional tasks and excluded the $x$-feature to study a dimensionality-agnostic version of MetaBO. Indeed, we evaluated the resulting neural AFs *without retraining* for dimensionalities $D \in \{3, 4, 5\}$. The results (Fig. 5) show that MetaBO is capable of learning neural AFs which perform better than or at least on on-par with EI on these general function classes.

## 6 CONCLUSION AND FUTURE WORK

We introduced MetaBO, a novel method for transfer learning in the framework of BO. Via a flexible meta-learning approach, we inject prior knowledge directly into the optimization strategy of BO using neural AFs. The experiments show that our method consistently outperforms existing methods, for instance in simulation-to-real settings or on hyperparameter search tasks. Our approach is broadly applicable to a wide range of practical problems, covering both the cases of scarse and abundant source data. The resulting neural AFs can represent search strategies which go far beyond the abilities of current approaches which often rely on weighted superpositions of priors over the optimization domain obtained from the source data with standard AFs. In future work, we aim to tackle the multi-task multi-fidelity setting (Valkov et al., 2018), where we expect MetaBO's sample efficiency to be of high impact.

### ACKNOWLEDGEMENTS

We want to thank Julia Vinogradska, Edgar Klenske, Aaron Klein, Matthias Feurer, Gerhard Neumann, as well as the anonymous reviewers for valuable remarks and discussions which greatly helped to improve this paper.

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

## A    ADDITIONAL EXPERIMENTAL RESULTS

### A.1    INTERPRETATION OF NEURAL AF SEARCH STRATEGIES

We provide additional experimental results to demonstrate that MetaBO's neural AFs learn representations that go beyond some kind of standard AF combined with a prior over $\mathcal{D}$.

**Emergence of Non-Greedy Search Strategies**    To obtain intuition about the kind of search strategies MetaBO is able to learn, we devised two classes of one-dimensional toy objective functions.

The first class of objective functions (Rhino-1, cf. Fig. 6) is generated by applying random translations sampled uniformly from $t \in [-0.2, 0.2]$ to a function which is given by the superposition of two Gaussian bumps with different heights and widths and fixed distance,

$$f_{\mathrm{R1}}(x, t) \equiv 0.5 \cdot \mathcal{N}(x \,|\, \mu = 0.3 - t, \sigma = 0.1) + 3.0 \cdot \mathcal{N}(x \,|\, \mu = 0.7 - t, \sigma = 0.01), \quad (6)$$

where we define $\mathcal{N}(x \,|\, \mu, \sigma) \equiv \exp(-1/2 \cdot (x - \mu)^2/\sigma^2)$. The second class of objective functions (Rhino-2, cf. Fig. 7) is given by uniformly sampling the parameter $h \in [0.6, 0.9]$ of the function

$$f_{\mathrm{R2}}(x, h) \equiv h \cdot \mathcal{N}(x \,|\, \mu = 0.2, \sigma = 0.1) + 2.0 \cdot \mathcal{N}(x \,|\, \mu = h, \sigma = 0.01) - 1.0. \quad (7)$$

For both of these function classes it is intuitively clear that the optimal search strategy involves a first non-greedy evaluation to identify the specific instance of the target function. Indeed, for all instances of these function classes, the smaller and wider bumps overlap and encode information about the position of the sharp global optimum. Therefore, an optimal strategy spends the first evaluation at a fixed position $x_0$ where all smaller and wider bumps have non-negligible heights $y_0$. Then, for both function classes, the global optimum $x^*$ can be determined exactly from $y_0$ (if we assume noiseless evaluations), such that $x^*$ can be found in the second step. Figs. 6, 7 show that MetaBO indeed learns such non-greedy optimization strategies, which go far beyond a simple combination of a prior over $\mathcal{D}$ with some kind of standard AF. As mentioned in the main part of this paper, we suppose that MetaBO employs similar strategies on more complex function classes. For instance, we observe in the experiments on the global optimization benchmark functions (Fig. 2) that MetaBO consistently starts with higher regret than the pre-informed TAF which suggests that it learned to spend a few non-greedy evaluations at the beginning of an optimization run to identify the specific instance of the target function.

**Additional Baseline Methods**    To provide further evidence that MetaBO's neural AFs learn representations that go beyond a simple prior over $\mathcal{D}$ combined with some kind of standard AF, we show results for two additional baseline AFs which rely on such a naive combination.

We define the AF GMM-UCB as the following convex combination of a Gaussian Mixture Model (GMM) and the standard AF UCB:

$$\mathrm{GMM} - \mathrm{UCB}(\boldsymbol{x}) \equiv w \cdot \mathrm{GMM}(\boldsymbol{x}) + (1 - w) \cdot \mathrm{UCB}(\boldsymbol{x}). \quad (8)$$

The GMM is defined to have $n_{\mathrm{comp}}$ components and is fitted to the best designs from each of the $M$ source tasks. Further, UCB is defined as

$$\mathrm{UCB}(\boldsymbol{x}) \equiv \mu(\boldsymbol{x}) + \beta\sigma(\boldsymbol{x}), \quad (9)$$

and we choose $\beta = 2$ as is common in BO.

Furthermore, we define EPS-GREEDY as the AF which in each optimization samples without replacement from the set of best designs of each of the source tasks step with probability $\epsilon$ and uses standard EI with probability $1 - \epsilon$.

Note that these baseline methods are similar in spirit to the TAF-approach evaluated in the main part of this paper. Indeed, TAF, GMM-UCB, and EPS-GREEDY all rely on some kind of prior over $\mathcal{D}$ determined using the source data which is combined through a weighted superposition with some standard AF. However, TAF uses more principled methods (TAF-ME, TAF-R) to adaptively determine the weights of this superposition.

To obtain optimal performance of GMM-UCB and EPS-GREEDY, we chose the parameters for these methods by grid search on the test set[7] w.r.t. the median simple regret summed from $t = 0$ to

---

[7]This yields an upper bound on the possible performance of GMM-UCB and EPS-GREEDY, as in practice one would have to estimate the parameters using a separate validation set.

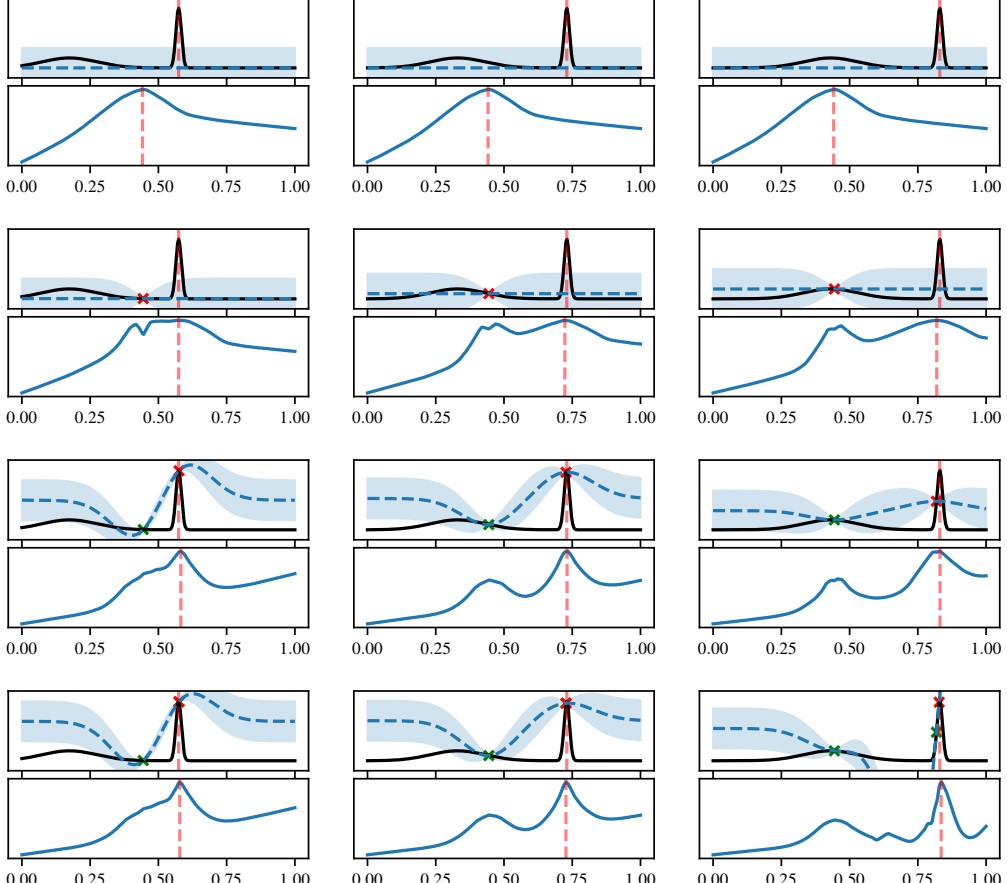

Figure 6: Visualization of three BO episodes with neural AFs on the $1D$ Rhino-1 task. Each column of this figure correspond to one episode with three optimization steps. The uppermost row corresponds to the prior state before the objective function was queried. The fourth row depicts the state after three evaluations. Each subfigure shows the GP mean (dashed blue line), GP standard deviation (blue shaded area), and the ground truth function (black) in the upper panel as well as the neural AF in the lower panel. Dashed red lines indicate the maxima of the ground truth function and of the neural AF. Red and green crosses indicate the recorded data (the red cross corresponds to the most recent data point). Each instance of this task is generated by randomly translating an objective function with two peaks of different heights and widths. The distance between the local and global optimum is the same for each instance. MetaBO learns a sophisticated sampling strategy, spending a non-greedy evaluation at the beginning of each episode at a position where the smaller but wider peaks overlap for every instance of the function class to gain information about the location of the global optimum. Using this strategy, MetaBO is able to find the global optimum very efficiently.

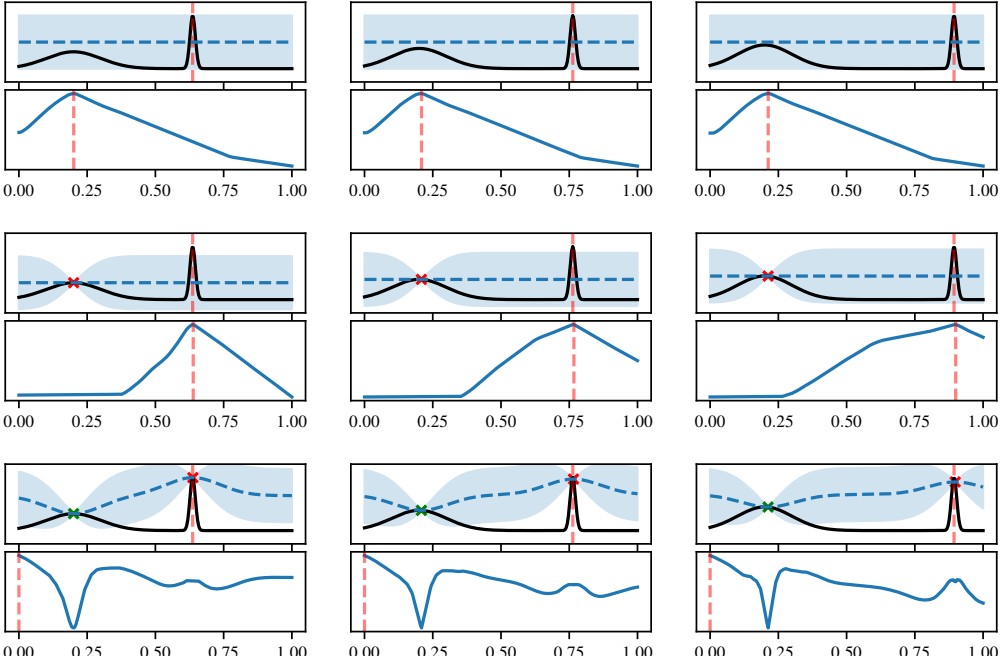

Figure 7: Visualization of three episodes from the $1D$ Rhino-2 task. Each column of this figure correspond to one episode with two optimization steps. The uppermost row corresponds to the prior state before the objective function was queried. The third row depicts the state after two evaluations. Each subfigure shows the GP mean (dashed blue line), GP standard deviation (blue shaded area), and the ground truth function (black) in the upper panel as well as the neural AF in the lower panel. Dashed red lines indicate the maxima of the ground truth function and of the neural AF. Red and green crosses indicate the recorded data (the red cross corresponds to the most recent data point). Each instance of this task is generated by sampling the height $h$ of a wide bump at a fixed location $x = 0.2$ and placing a sharp peak at $x = h$. MetaBO learns a sophisticated sampling strategy, spending a non-greedy evaluation at $x \approx 0.2$ at the beggining of each episode to gain information about the location of the global optimum. Using this strategy, MetaBO is able to find the global optimum very efficiently.

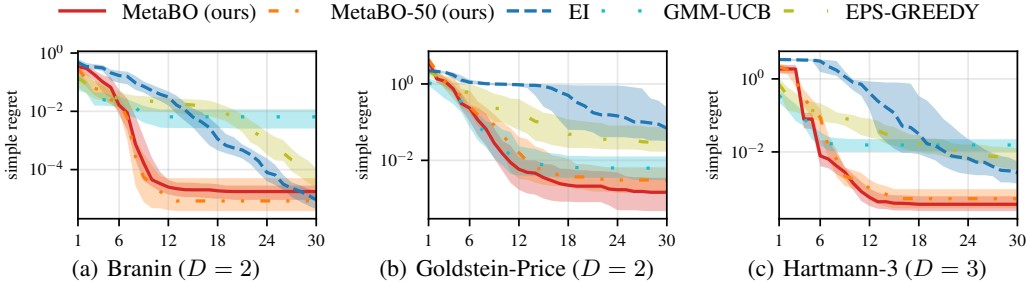

(a) Branin ($D = 2$)  (b) Goldstein-Price ($D = 2$)  (c) Hartmann-3 ($D = 3$)

Figure 8: Performance on three global optimization benchmark functions with random translations sampled uniformly from $[-0.1, 0.1]^D$ and scalings from $[0.9, 1.1]$. We present results for two additional baseline methods (GMM-UCB, EPS-GREEDY) which rely on a weighted superposition of a prior over $\mathcal{D}$ obtained from $M = 50$ source tasks and a standard AF and can thus be easily interpreted. As MetaBO produces more sophisticated search strategies, these approaches are not able to surpass MetaBO's performance.

Table 2: Optimal parameters of GMM-UCB and EPS-GREEDY (determined on the test set).

|  | $w$ | $n_{\text{comps}}$ | $\epsilon$ |
|---|---|---|---|
| **Branin** | 0.22 | 3 | linear schedule |
| **Goldstein-Price** | 0.22 | 1 | 0.55 |
| **Hartmann-3** | 0.11 | 2 | linear schedule |

$t = T = 30$. To tune $w$ for GMM-UCB we tested 10 linearly spaced points in $[0.0, 1.0]$ as well as a schedule which reduces $w$ from 1.0 to 0.0 over the course of one episode. Furthermore, we tested numbers of GMM-components $n_{\text{comp}} \in \{1, 2, 3, 4, 5\}$. Similarly, for EPS-GREEDY we tested $\epsilon$ on 10 linearly spaced points in $[0.0, 1.0]$ and also evaluated a schedule which reduces $\epsilon$ from 1.0 to 0.0 over an episode.

In Fig. 8 we display the performance of GMM-UCB and EPS-GREEDY on the global optimization benchmark functions Branin, Goldstein-Price, and Hartmann-3 with the optimal parameter configurations (cf. Tab. 2) and with $M = 50$ source tasks. MetaBO outperforms both GMM-UCB and EPS-GREEDY which provides additional evidence that neural AFs learn representations which go beyond a simple combination of standard AFs with a prior over $\mathcal{D}$.

## A.2 Dependence on the Number of Source Tasks

We argued in the main part of this paper that one main advantage of MetaBO over existing transfer learning methods for BO is its ability to process a very large amount of source data because it does not store all available data in GP models (in contrast to TAF) but rather accumulates the data in the neural AF weights. For tasks where source data is abundant (e.g., when it comes from simulations, cf. Fig. 3), this frees the user from having to select a small subset of representative source tasks by hand, which can be intricate or even impossible for complex tasks. In addition, we showed in our experiments that MetaBO's applicability is not restricted to such cases, but that it also performs favourably with the same amount of source data as presented to the baseline methods on tasks which do not require a very large amount of source data to be solved efficiently (cf. Figs. 2, 4).

In Fig. 9 we provide further evidence for this aspect by plotting the performance of MetaBO for different numbers $M$ of source tasks on the Branin function and on functions from the simulation of the Furuta pendulum stabilization task. The results indicate that on the Branin function a small number of source tasks is already sufficient to obtain strong optimization performance. In contrast, the more complex stabilization task requires a much larger amount of source data to be solved reliably.

We emphasize that MetaBO's evaluation runtime does not depend on the number $M$ of source tasks because a neural AF evaluation only requires one forward pass through a neural AF of fixed size. Therefore, it scales well to the regime of abundant source data. In contrast, TAF-ME's runtime scales

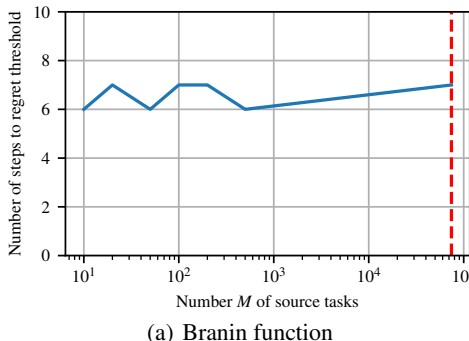
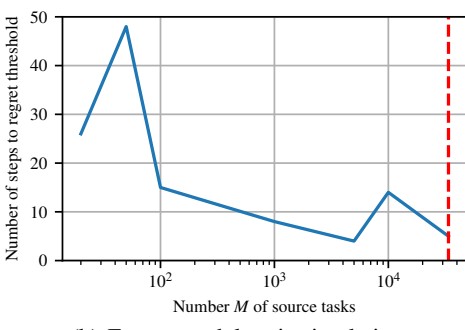

(a) Branin function              (b) Furuta pendulum in simulation

Figure 9: Dependence of MetaBO's performance on the number of source tasks provided during training on the Branin function (cf. Fig. 2(a)) and on the stabilization task for the Furuta pendulum in simulation (cf. Fig. 3(a)). We show the number of steps MetaBO requires to reach a given performance in terms of median regret over 100 test functions in dependence of the number $M$ of source tasks. As in the main part of this paper, we chose a constant budget of $T = 30$ on the Branin function and of $T = 50$ on the stabilization task. The dashed red line indicates the number of source tasks seen by the full version of MetaBO (a new function is sampled from the training distribution at the beginning of each optimization episode) at the point of convergence of meta-training. For the Branin function we chose the regret threshold $R = 10^{-3}$, which corresponds to the median final performance of TAF after $t = 30$ steps as presented in the main part of this paper (Fig. 2(a)). For the Furuta stabilization task, we chose the regret threshold $R = 1.0$, which corresponds approximately to the regret that has to be reached in simulation to allow stabilization on the real system. The results show that on the Branin function already a small number of source tasks is enough to obtain a powerful optimization strategy. In contrast, neural AFs trained on the more complex simulation-to-real task benefit from MetaBO's ability to process a very large amount of source tasks.

Table 3: Comparison of evaluation runtimes per BO episode with budget $T = 30$ in s for various AFs, averaged over 10 BO runs. We show MetaBO's runtime for $M = 50$ source tasks as well as for the full version (where a new function is sampled from the training distribution in each BO run). For TAF, we indicate $M$ and the number $N$ of data points per source task by TAF-ME-$M$-$N$ and TAF-R-$M$-$N$. Note that the absolute figures of the reported runtimes obviously depend on the hardware architecture used for the evaluation.

|  | **Branin** | **Goldstein-Price** | **Hartmann-3** |
|---|---|---|---|
| **EI** | 0.13 | 0.13 | 0.16 |
| **MetaBO-50** | 0.60 | 0.55 | 0.82 |
| **MetaBO-full** | 0.62 | 0.59 | 0.81 |
| **TAF-ME-50-100** | 13 | 14 | 24 |
| **TAF-ME-50-200** | 29 | 28 | 35 |
| **TAF-ME-100-100** | 17 | 19 | 30 |
| **TAF-ME-100-200** | 47 | 49 | 65 |
| **TAF-R-50-100** | 50 | 50 | 56 |
| **TAF-R-50-200** | 61 | 60 | 69 |
| **TAF-R-100-100** | 100 | 100 | 110 |
| **TAF-R-100-200** | 120 | 120 | 140 |

linearly in the number $M$ of source tasks and quadratically in the number $N$ of data points per source task, while TAF-R shows an even stronger dependence on $M$ due to the computation of the pairwise ranks. We underline this scaling behavior by presenting measured evaluation runtimes in Tab. 3.

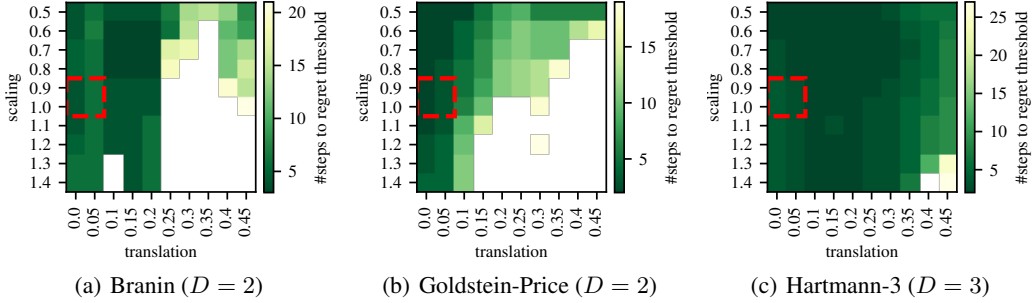

(a) Branin ($D = 2$)    (b) Goldstein-Price ($D = 2$)    (c) Hartmann-3 ($D = 3$)

Figure 10: Generalization of neural AFs to functions outside of the training distribution (translations $t \in [-0.1, 0.1]$, scalings $s \in [0.9, 1.1]$, red square) on Branin, Goldstein-Price, and Hartmann-3. We evaluated the neural AFs on 100 test distributions with disjoint ranges of translations and scalings, each corresponding to one tile of the heatmap. The $x$- and $y$-labels of each tile denote the lower bounds of the translations $t$ and scalings $s$ of the respective test distribution from which the parameters were sampled uniformly (for each dimension we sampled the translation and its sign independently). The color encodes the number of optimization steps required to reach a given regret threshold. White tiles indicate that this threshold could not be reached withtin $T = 30$ optimization steps. The regret threshold was fixed for each function separately: we set it to the $1\%$-percentile of the set of regrets corresponding to function evaluations on a Sobol grid of one million points in the domain of the original objective functions.

### A.3  GENERALIZATION BEHAVIOR

As described in the main part of this paper, MetaBO's primary use case is transfer learning, i.e., to speed up optimization on target functions similar to the source objective functions. Put differently, we are mainly interested in MetaBO's performance on unseen functions drawn from the training distribution. Nevertheless, studying MetaBO's generalization performance to functions outside of the training distribution can give interesting insights into the nature of the tasks we considered in the main part. Therefore, we present a study of MetaBO's generalization performance on the global optimization benchmark functions (Fig. 10) as well as on the simulation-to-real experiment (Fig. 11).

The results on the simulation-to-real task show that the neural AF generalizes better to heavy and long than to lightweight and short pendula. We suppose that this result is related to the fact that lightweight and short pendula show much faster dynamics due to their small moments of inertia than heavier and longer ones and are thus much harder to stabilize. Put more precisely, the change of the optimization landscape is much more pronounced when moving to lighter and smaller pendula than in the other direction. Similar conclusions can be drawn for the translated and scaled global optimization benchmark functions.

### A.4  FULL SET OF RESULTS FROM MAIN PART

**Global Optimization Benchmark Functions**    We provide the full set of results for the experiment on the global optimization benchmark functions. In Fig. 12 we also include results for TAF with $\mathrm{M} = 20$, showing that TAF's performance does not necessarily increase with more source data.

**Simulation-to-Real Experiment**    We provide the full set of results for the experiment on the global optimization benchmark functions, including the results for TAF-50, cf. Fig. 13.

## B  EXPERIMENTAL DETAILS

To foster reproducibility, we provide a detailed explanation of the settings used in our experiments and make source code available online.[8]

---

[8] https://github.com/boschresearch/MetaBO

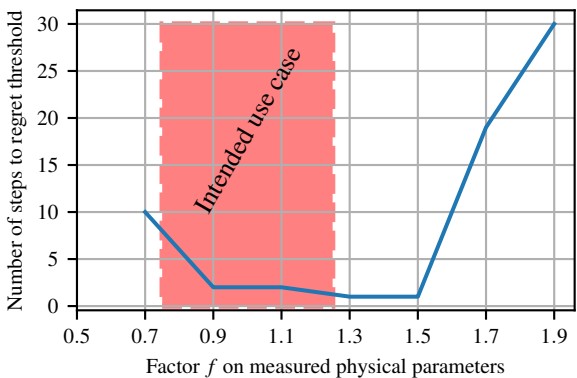

Figure 11: Generalization of neural AFs to functions outside of the training distribution (75% to 125% of measured physical parameters, red square) on the simulation-to-real task. We evaluated neural AFs on test distributions with disjoint ranges of physical parameters (masses and lengths of the pendulum and arm). We sampled each physical parameter $p_i$ uniformly on $[f \cdot p_{i,\text{measured}}, (f + 0.2) \cdot p_{i,\text{measured}}]$. Therefore, $f = 0.9$ corresponds to the interval containing the measured parameters. We plot $f$ on the $x$-axis and the number of steps required to reach a regret threshold of $R = 1.0$ on the $y$-axis. Following our experience, this corresponds approximately to the regret that has to be reached in simulation to allow stabilization on the real system. We emphasize that the intended use case of MetaBO is on systems inside of the training distribution marked in red, as this distribution is chosen such that the true parameters are located inside of it with high confidence when taking into account the measurement uncertainty. Note that for small $f$ the system becomes very hard to stabilize (lightweight and short pendula) such that the optimization landscape differs significantly from the training distribution, which is why the regret threshold cannot be reached within 30 steps for $f \leq 0.5$.

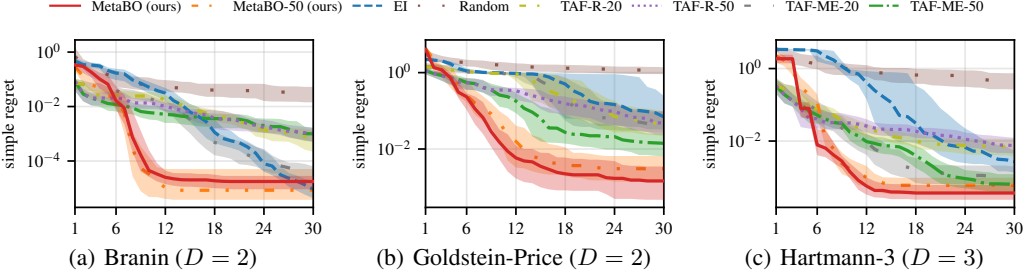

(a) Branin ($D = 2$)          (b) Goldstein-Price ($D = 2$)          (c) Hartmann-3 ($D = 3$)

Figure 12: Performance on three global optimization benchmark functions with random translations sampled uniformly from $[-0.1, 0.1]^D$ and scalings from $[0.9, 1.1]$. To test TAF's performance, we randomly picked $M$ source tasks from this function class and evaluated both the ranking-based version (TAF-R-$M$) and the mixture-of-experts version (TAF-ME-$M$). We show results for $M \in \{20, 50\}$. Note that TAF's performance does not necessarily increase with more source data. We trained MetaBO on the same set of source tasks as TAF-50 (MetaBO-50). In contrast to TAF, MetaBO can also be trained without manually restricting the set of available source tasks. The corresponding results are labelled "MetaBO". MetaBO outperformed EI by clear margin, especially in early stages of the optimization. After few steps used to identify the specific instance of the objective function, MetaBO also outperforms both flavors of TAF over wide ranges of the optimization budget.

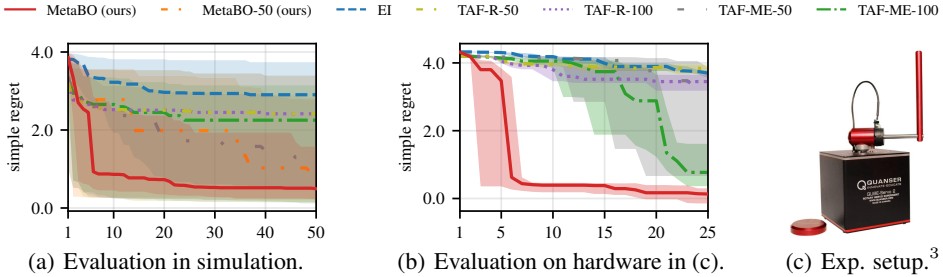

(a) Evaluation in simulation.   (b) Evaluation on hardware in (c).   (c) Exp. setup.[3]

Figure 13: Performance on a simulation-to-real task (cf. text). MetaBO and TAF used source data from a cheap numerical simulation. (a) Performance on an extended training set in simulation. (b) Transfer to the hardware depicted in (c), averaged over ten BO runs. MetaBO learned robust neural AFs with very strong early-time performance and online adaption to the target objectives, which reliably yielded stabilizing controllers after less than ten BO iterations while TAF-ME-50, TAF-ME-100, TAF-R-50, TAF-R-100, and EI explore too heavily. Comparing the results for MetaBO and MetaBO-50 in simulation, we observe that MetaBO benefits from its ability to learn from the whole set of available source data, while TAF's applicability is restricted to a comparably small number of source tasks.

### B.1 General Implementation Details

In what follows, we explain all hyperparameters used in our experiments and summarize them in Tab. 4. We emphasize that we used the same MetaBO hyperparameters for all our experiments, making our method easily applicable in practice.

**Gaussian Process Surrogate Models** We used the implementation GPy (GPy, 2012) with squared-exponential kernels (Matern-5/2 kernels for the corresponding experiments on general function classes) with automatic relevance determination and a Gaussian noise model and tuned the corresponding hyperparameters (noise variance, kernel lengthscales, kernel signal variance) offline by fitting a GP to the objective functions in the training and test sets using type-2 maximum likelihood. We also used the resulting hyperparameters for the source GPs of TAF. We emphasize that our method is fully compatible with other (online) hyperparameter optimization techniques, which we did not use in our experiments to arrive at a consistent and fair comparison with as few confounding factors as possible.

**Baseline AFs** As is standard, we used the parameter-free version of EI. For TAF, we follow Wistuba et al. (2018) and evaluate both the ranking-based (TAF-R) as well as the product-of-experts (TAF-ME) versions. We detail the specific choices for the number of source tasks $M$ and the number of datapoints $N_{\mathrm{TAF}}$ contained in each source GP in the main part of this paper.

For EI we used the midpoint of the optimization domain $\mathcal{D}$ as initial design. For TAF we did not use an initial design as it utilizes the information contained in the source tasks to warmstart BO. Note that MetaBO also works without any initial design.

**Maximization of the AFs** Our method is fully compatible with any state-of-the-art method for maximizing AFs. In particular our neural AFs can be optimized using gradient-based techniques. We chose to switch off any confounding factors related to AF maximization and used a hierarchical gridding approach for all evaluations as well as during training of MetaBO. For the experiments with continuous domains $\mathcal{D}$, i.e. all experiments except the HPO task, we first put a multistart Sobol grid with $N_{\mathrm{MS}}$ points over the whole optimization domain and evaluated the AF on this grid. Afterwards, we implemented local searches from the $k$ maximal evaluations via centering $k$ local Sobol grids with $N_{\mathrm{LS}}$ points, each spanning approximately one "unit cell" of the multistart grid, around the $k$ maximal evaluations. The AF maximum is taken to be the maximal evaluation of the AF on these $k$ Sobol grids. For the HPO task, the AF maximum can be determined exactly because the domain is discrete.

**Reinforcement Learning Method** We use the trust-region policy gradient method Proximal Policy Optimization (PPO) (Schulman et al., 2017) as the algorithm to train the neural AF.

Table 4: Parameters of the MetaBO framework used in our experiments.

| Description | Value in experiments |
|---|---|
| *BO/AF parameters* | |
| Cardinality $N_{\mathrm{MS}}$ of multistart grid | |
|     Branin, Goldstein-Price | 1000 |
|     Hartmann-3 | 2000 |
|     Simulation-to-real | 10000 |
|     GPs ($D = 1, 2, 3, 4, 5$) | 500, 1000, 2000, 3000, 4000 |
| Cardinality $N_{\mathrm{LS}}$ of local search grid | $N_{\mathrm{MS}}$ |
| Number $k$ of multistarts | 5 |
| *MetaBO parameters* | |
| Cardinality of $\boldsymbol{\xi}_{\mathrm{global}}$ | $N_{\mathrm{MS}}$ |
| Cardinality of $\boldsymbol{\xi}_{\mathrm{local},t}$ | $k$ |
| Neural AF architecture | 200 - 200 - 200 - 200, relu activations |
| *PPO parameters* (Schulman et al., 2017) | |
| Batch size | 1200 |
| Number of epochs | 4 |
| Number of minibatches | 20 |
| Adam learning rate | $1 \cdot 10^{-4}$ |
| CPI-loss clipping parameter | 0.15 |
| Value network architecture | 200 - 200 - 200 - 200, relu activations |
| Value coefficient in loss function | 1.0 |
| Entropy coefficient in loss function | 0.01 |
| Discount factor $\gamma$ | 0.98 |
| GAE-$\lambda$ (Schulman et al., 2015) | 0.98 |

**Reward Function**  If the true maximum of the objective functions is not known at training time, we compute $R_t$ with respect to an approximate maximum and define the reward to be given by $r_t \equiv -R_t$. This is the case for the experiment on general function classes (GP samples) where we used grid search to approximate the maximum as well as for the simulation-to-real task on the Furuta pendulum where we used the performance of a linear quadratic regulator (LQR) controller as an approximate maximum. For the experiments on the global optimization benchmark functions as well as on the HPO tasks, we do know the exact value of the global optimum. In these cases, we use a logarithmic transformation of the simple regret, i.e., $r_t \equiv -\log_{10} R_t$ as the reward signal. Note that we also consistently plot the logarithmic simple regret in our evaluations for these cases.

**Neural AF Architecture**  We used multi-layer perceptrons with relu-activation functions and four hidden layers with 200 units each to represent the neural AFs.

**Value Function Network**  To reduce the variance of the gradient estimates for PPO, a value function $V_\pi(s_t)$, i.e., an estimator for the expected cumulative reward from state $s_t$, can be employed (Schulman et al., 2015). In this context, the optimization step $t$ and the budget $T$ are particularly informative features, as for a given sampling strategy on a given function class they allow quite reliable predictions of future regrets. Thus, we propose to use a separate neural network to learn a value function of the form $V_\pi(s_t) = V_\pi(t, T)$. We used an MLP with relu-activations and four hidden layers with 200 units each for the value network.

**Computation Time**  For training MetaBO, we employed ten parallel CPU-workers to record the data batches and one GPU to perform the policy updates. Depending on the complexity of the objective function evaluations, training a neural AF for a given function class took between approximately $30\,\mathrm{min}$ and $10\,\mathrm{h}$ on this moderately complex architecture.

