# OpenReview forum: "Meta-Learning Acquisition Functions for Transfer Learning in Bayesian Optimization"
_ICLR.cc/2020/Conference — Accept (Spotlight)_

### Official Review · AnonReviewer2 · 2019-10-16
**Official Blind Review #2**

**Rating:** 8

**Review:**

The authors present MetaBO, which uses reinforcement learning to meta-learn the acquisition function (AF) for Bayesian Optimization (BO) instead of using a standard constant AF. The authors shows that MetaBO enables transferring knowledge between tasks and increasing sample efficiency on new tasks. The paper is mostly clearly written and I am not aware of existing work on meta-learning the AF for BO. However, the approach is related to Chen et al, which is cited in the text but not used as a baseline. It is also not shown clearly enough how the performance of MetaBO depends on the number of training tasks and distance between training and test tasks. I therefore consider the paper as borderline.

Major comments
=============
1. The presented approach is very similar to Chen et al, which is discussed in the related work section but not used as a baseline. Although Chen et al assumed that f(x) is differentiable, their approach can be easily generalized to non-differentiable functions by using RL as Chen et al discussed in the last paragraph of section 2.1. Chen et al does not depend on a GP and is therefore more scalable. The source code is publicly available (https://github.com/deepmind/learning-to-learn) and you can also adapt your implementation by removing the GP part.

2. Global Optimization Benchmark Functions: How does the performance of MetaBO depend on the number of training samples (number of training tasks times the budget T)?

3. Figure 3: How does MetaBO generalizes to functions that are translated and scaled at the same time? This can be visualized as a heatmap with the scaling and translation on the x and y axis, and using the color to show the number of steps to reach a certain reward. How does the generalization performance depend on the noise level, where the noise can be sampled from standard normal distribution? Why does EI perform better if the function is translated more?

4. Simulation-to-Real task: How does the generalization performance of MetaBO depend on the distance between training and source tasks (x-axis: distance; y-axis: steps to reach a certain reward)? You sampled test tasks 10%-200% around the true parameters. Test tasks can therefore have identical or similar parameters than training tasks.

5. Simulation-to-Real task: How does the performance depend on the number of training tasks (x-axis: # training tasks; y-axis: steps to reach a certain performance)?

Minor comments
=============
6. Section 1, 2nd paragraph: The performance of BO also depends on the GP kernel and kernel hyper-parameters, not only the AF. Please mention this. Similarly, ‘no need to calibrate any hyperparameter’ in section 4 ignores GP hyper-parameters. Please clarify.

7. Section 2, 4th paragraph: A Neural Process (https://arxiv.org/abs/1807.01622) is another scalable alternative to a GP. Please cite.

8. Section 3, 2nd paragraph: Please cite standard AFs such as EI, PI, UCP.

9. Section 4, last paragraph before ‘Training procedure’. The state s_t is undefined at this point. This section misses a clear description of the state, reward, and transition function of the MDB. Does the state s_t take previous function evaluations into account (e.g. via a RNN state), or only \mu and \sigma at the current step t? Does the state include the time step as described in the text and in the section about the value network in the appendix but not in table 1.

10. Section 4, ‘the state corresponds to the entire functions’.  It only depends on the first two moments (and the time step t?).

11. Section 4: replace ‘not to be available’ by ‘unavailable’.

12. Section 4: reference or describe ‘Sobol grid’.

13. Section 4: The approach to maximize the AF on grid points does not scale to high-dimensional search spaces. Please also clarify how global and local grid points were chosen. In particular, ‘local maximization’ is unclear. Also, ‘cheap approximation’ of the global maximum of f(x) is infeasible if the search space is high-dimensional.

14. Please move figure 3 above figure 4.

**Experience Assessment:**

I have published one or two papers in this area.

**Review Assessment: Checking Correctness Of Derivations And Theory:**

N/A

**Review Assessment: Checking Correctness Of Experiments:**

I carefully checked the experiments.

**Review Assessment: Thoroughness In Paper Reading:**

I read the paper thoroughly.

---

> ### Author Response · Authors · 2019-11-14
> **Author response to official blind review #2**
>
> We are grateful for your detailed comments and suggestions to further improve our paper. We hope that the following remarks and the updated version of our paper help to improve your impression of MetaBO. We highlighted the updated sections in yellow in the updated PDF.
>
> 1.) Comparison with Chen et al. [1]
> -----------------------------------
> We agree that Chen et al., "Learning to learn *without* gradient descent by gradient descent" [1], would be an interesting baseline for our method. We would like to clarify why we did not benchmark against this approach.
>  - There is no code available for this method. The link provided by the reviewer points to a repository for a *different paper with a similar name* (namely Andrychowicz et al., "Learning to learn *by* gradient descent by gradient descent" [2]). This tackles the problem of learning *local, gradient-based* optimization and is thus not applicable in our scope of *global, derivative-free* optimization. We have exchanged several emails with the first author of [1] (Yutian Chen) about availability of the code (already before the time of our submission), and when we emailed again this week based on the reviews, he kindly allowed us to quote his email answer to our question whether the code for "Learning to learn without gradient by gradient descent" could be made publicly available: "Unfortunately I haven't been able to open source code due to lack of time. I'll check if there's some way of sharing part of the code with you, but I can't guarantee on that.", Yutian Chen, Sept. 24th, 2019.
>  - We spent considerable effort trying to reproduce the results on our own. However, we were not able to reach the performance reported in the paper's supervised setting. Therefore, unfortunately, an adaptation of the proposed method to our transfer learning setting, which would require an even more complex RL-approach (due to the lack of gradients of the objective functions, as you correctly pointed), seems out of reach for us at the moment. In fact, this is one of the reasons why we chose to retain the GP-surrogate model and only tackle the less complex problem of learning solely the AF.
>
> 2.) Investigation of generalization performance
> -----------------------------------------------
> As suggested, we extended and improved the experiments assessing MetaBO's generalization performance (App. A.2).
>  - We included scalings and translations in our experiments on the generalization performance of MetaBO on the global optimization benchmark functions. We present the proposed heatmap visualization in the updated PDF (App. A.2, Fig. 8).
>  - We also present the generalization performance on the simulation to real task (App. A.2, Fig. 9).
>  - We now *do not include* the training distribution as a subset of the test distribution anymore in these experiments. We emphasize however that the intended use case of our method is an evaluation on functions from the training distribution. In the simulation-to-experiment task, for example, the training distribution is constructed on a range around measured physical parameters which is chosen such that the true parameters of the hardware system lie in this range with high confidence. Nevertheless, these experiments can give interesting insights into the nature of the tasks and we gladly add them to the paper.
>
> 3.) Dependence on the number of source tasks
> --------------------------------------------
> As you suggested, we extended our experiments on the dependence of MetaBO's performance on the number of training tasks (App. A.3, Fig. 10). The new experiments underline again that MetaBO performs favorably both in the regime of scarse and abundant source data. Furthermore, MetaBO scales much better to large amount of source tasks than the baseline methods (App. A.3, Tab. 3).
>
> 4.) Minor comments
> ------------------
>  - We reorganized and improved Sec. 4 explaining our method and hope that this helps to clarify your questions.
>  - We added the proposed references.
>  - We corrected some minor spelling mistakes.
>
> References:
> [1] Chen et al., "Learning to learn without gradient descent by gradient descent", ICML 2017
> [2] Andrychowicz et al., "Learning to learn by gradient descent by gradient descent", NIPS 2016

---

### Official Review · AnonReviewer3 · 2019-10-18
**Official Blind Review #3**

**Rating:** 6

**Review:**

Summary: The authors propose a meta-learning based alternative to standard acquisition functions (AFs), whereby a pretrained neural network outputs acquisition values as a function of hand-chosen features. These neural acquisition functions (NAFs) are trained on sets of related tasks using standard RL methods and, subsequently, employed as drop-in replacements for vanilla AFs at test-time.


Feedback:
Overall, the proposed method makes sense and would benefit from further experimental ablation. Using RL to automatically derive (N)AFs is a nice change of pace from the hand-crafted heuristics that dominate BO. I like the ideas at play here and hope that you will convince me to amend my score.

Results on synthetic functions presented in the body of the paper demonstrate that NAF outperforms, e.g., EI when transferring between homogenous tasks. In contrast, results when transferring between relatively heterogenous functions (Fig. 9) indicate that the aforementioned performance gain reflect NAFs ability to specialize. Two things remain unclear however:
    a. What types of regularity are NAFs able to exploit?
    b. How quickly do NAFs benefits fall off as tasks become increasingly heterogenous?


Regarding (a), I am not yet convinced that NAFs learn representations that go beyond standard AFs combined with a prior over $x$. To help test this hypothesis, here is a sketch of a simple baseline algorithm:
  1. Fit, e.g., a Gaussian Mixture Model to the top $k=1$ designs $x^{*}_{i}$ on observed tasks $i \in [1, N]$,
  2. Given a new task $f_{j}$, let log-likelihood $GMM(x)$ act as a 'prior' of sorts on $x$
  3. Use cross-validation to tune the scalar parameter $w$ of a new AF defined as the convex combination:

        GMM-UCB(x_{k}) = w * GMM(x_{k}) + (1 - w) * UCB(x_{k})
                                       = w * GMM(x_{k}) + (1 - w) * [\mu_{k} + \sqrt{\beta} * \sigma_{k}].

I suggest using UCB both because NAF could easily learn it from its inputs and because EI values often decay dramatically over the course of BO (I usually set UCB's confidence parameter to a fixed value $\beta = 2$).

Further simplifying this idea, you could instead use an $\epsilon$-greedy style heuristic that, with probability $\epsilon$, samples without replacement from the set of historical minimizers and otherwise uses a standard AF. These baselines are comparatively straightforward and easily interpreted, so I hope that you will consider adding something along these lines.


Additionally, here are some questions/suggests to help probe (a-b):
  1. Another baseline: EI with multi-task GP? The cubic scaling should be fine for, e.g., 'xxx-20' multi-task variants.
  2. Extend experiments on functions drawn from GP priors (Fig 9):
      i. How does homogeneity (as enforced via the GP hyperprior) impact performance when transferring knowledge?
      ii. Rate of convergence suggests sampled tasks may be too easy; consider using Matern-5/2 and smaller lengthscales [*].
  3. What happens if you expand the task augmentation process to further include, e.g., flips and rotations?
  4. How do 'dimension-agnostic' versions of NAF (where $x$ is excluded from its input) perform on other synthetic tasks?
  5. Visualizing NAF (or the search strategies it produces) would be useful for building intuition.
  6. How were NAF input features chosen? Were alternatives such as also passing the 'best seen' value, considered?
  7. How easy to use are NAFs in comparison to alternative AFs (both in terms of training and test-time maximization)?
  8. Please report regret in log-scale (in appendix); currently, it is hard to tell what is going on in some places. Similarly, the tracked regret level in Figures 3 & 7 changes between tasks without explanation?


In summary, I genuinely want NAF to succeed but am not yet convinced of its performance. If you can provide empirical results to help extinguish my doubts, I will gladly change my assessment.


Nitpicks, Spelling, & Grammar:
  - Some minor spelling and/or grammatical error, but the paper reads fairly well.
  - On [Chen et al., 2017]:  To the best of my knowledge, these RNN-based methods only require the gradient of the loss function. For example, using GP-based EI as the training signal only requires differentiating through EI + GP rather than through the target function $f$. Similarly, in cases where gradients are not available, the authors elude to use of RL algorithms such as REINFORCE.

[*] For Matern-5/2, just change the prior on your basis functions' weight parameters (https://github.com/metabo-iclr2020/MetaBO/blob/master/metabo/environment/objectives.py#L295) from standard normal to multivariate-t with 5 degrees of freedom.

**Experience Assessment:**

I have published one or two papers in this area.

**Review Assessment: Checking Correctness Of Derivations And Theory:**

N/A

**Review Assessment: Checking Correctness Of Experiments:**

I carefully checked the experiments.

**Review Assessment: Thoroughness In Paper Reading:**

I read the paper thoroughly.

---

> ### Author Response · Authors · 2019-11-14
> **Author response 2/2 to official blind review #3**
>
> *** Second part of author's response, please also refer to first part ***
>
> 4.) Tracked regret level in generalization experiments
> ------------------------------------------------------
> Following suggestions of AnonReviewer2, we extended the experiments investigating MetaBO's generalization performance on the global optimization benchmark functions (App. A.2, Fig. 8). In these new experiments, we consistently used the 1%-percentile of evaluations of the respective objective functions on a Sobol grid with one million points as the regret threshold. (We used different regret thresholds for different functions, since an error of 0.1 might be a lot for one function but only little for another; the 1%-percentile does not suffer from this issue, as it does already adapt to the range of outputs of the function at hand.)
>
> 5.) Multi-task GPs
> ------------------
> We did not consider multi-task GPs as proposed by Swersky et al. [2] as a baseline method in our paper because it is reported in the literature [3] that the performance of this method degrades for more than approximately M=5 tasks. While it is indeed correct that MTBO's global probabilistic model should scale to M=20 tasks with a few tens of data points each, it is reported to be infeasible to "correctly" determine (using MCMC) the MxM parameters of the task-correlation kernel.
>
> 6.) Incumbent as input feature
> ------------------------------
> We thank you for the suggestion to add the incumbent to the set of input features of NAF. We will consider adding this feature in future experiments, but we note that this should only improve the performance of our method.
>
> 7.) Include flips and rotations
> -------------------------------
> We agree that this would be one of many further interesting experiments to perform. However, due to time constraints in the rebuttal phase, we decided to focus on your other suggestions, as we feel they might give a better impression of MetaBO's capabilities.
>
> 8.) Nitpicks, spelling, and grammar
> -----------------------------------
> - We went through the paper again and corrected some minor spelling and grammatic mistakes.
> - The loss function of Chen et al. [1] consists of the sum of the losses incurred over the optimization episodes performed during training. To be able to train this in a supervised fashion, one has to backpropagate gradients through the whole optimization episode which includes the objective function evaluations, the GP, as well as EI. Therefore, in the original setting of [1], gradients of the objective functions are necessary. It is indeed correct that Chen et al. used samples from a GP-prior as objective functions during training. However, to apply their method in a transfer learning setting, one would have to use the available source objective functions as the training distribution. Therefore, the gradients of these objectives would have to be available. Please refer also to our discussion of this point in the answer to AnonReviewer2's review.
>
> References:
> [1] Chen et al., "Learning to learn without gradient descent by gradient descent", ICML 2017
> [2] Swersky et al., "Multi-task bayesian optimization", NIPS 2013
> [3] Klein et al., "Fast bayesian optimization of machine learning hyperparameters on large datasets", AISTATS 2017

---

> ### Author Response · Authors · 2019-11-14
> **Author response 1/2 to official blind review #3**
>
> We thank you for the overall positive feedback and are grateful for numerous and detailed suggestions to improve our paper. We hope that the following remarks and the new results and clarifications in the updated version of the paper remedy your concerns and can convince you to amend your score. We highlighted the updated sections in yellow in the updated PDF.
>
> 1.) Suggested baseline methods, Visualizing MetaBO's search strategies
> ----------------------------------------------------------------------
> We gladly followed your suggestion to provide additional experiments to gain insight into the search strategies MetaBO produces. Please refer to App. A.1 for details.
>  - We implemented the two suggested baseline methods (GMM-UCB, eps-greedy) to determine whether MetaBO learns representations that go beyond standard AFs combined with a prior over x. To obtain an upper bound on what could be achieved by tuning the parameters w (of GMM-UCB) as well as \epsilon (of eps-greedy), we selected their best value on the test set (of course, this can't be done in practice, but even when this approach is allowed to "cheat" like this, MetaBO still performs better), cf. App. A.1, Tab. 2.
> On top of your suggestion, we also additionally considered a schedule which gradually decreases w and \epsilon over the course of an optimization episode in order to reduce the impact of the prior as data on the target task becomes more and more reliable (we saw this as the natural extension of your proposed baselines). MetaBO still outperforms the proposed baseline methods, indicating that it learns search strategies which go beyond a prior over x combined with standard AFs, cf. App. A.1, Fig. 7.
>  - We would like to point out that this was to be expected at least for GMM-UCB as this method is very closely related to the TAF-approach which served as a baseline in our paper. Indeed, TAF is also a weighted superposition of a prior from the source tasks (observed improvement according to the source GPs) and a standard AF (EI) on the target task. Moreover, TAF employs principled mechanisms to adjust the weights of this superposition according to the relevance of source data on the target tasks (resulting in the presented versions TAF-ME and TAF-RANKING).
>  - To shed further light on the search strategies MetaBO produces, we devised two simple one-dimensional toy problems (Rhino-1 (App. A.1, Fig. 5), Rhino-2 (App. A.1, Fig. 6)) to demonstrate that MetaBO learns to use non-greedy evaluations in the beginning of an episode to obtain high information gain (rather than low regret) about the target function. This results in more efficient search strategies compared to approaches which simply favour specific zones in the search space.
>
> 2.) Extended experiments on functions drawn from GP priors, dimensionality-agnostic NAFs
> ----------------------------------------------------------------------------------------
> As suggested, we extended and improved the experiments on objective functions sampled from GP priors. Please refer to App. A.4 for details.
>  - To increase the complexity of the tasks, we performed experiments on smaller lengthscales with RBF-kernel and additionally performed experiments using the Matern-5/2 kernel (App. A.4, Fig. 11).
>  - We would like to emphasize that the experiments on GP priors merely serve as a sanity check in our paper. The focus of our work has been the transfer-learning setting in which the x-feature plays a central role as it enables MetaBO to recognize structure in the source tasks to learn sophisticated sampling strategies (as exemplified by our new Rhino-experiments). Note that all other considered transfer learning methods (including GMM-UCB, eps-greedy and TAF) also rely on this input feature (through the GMM, the best source designs, and the source GPs, respectively). Therefore, we did not further investigate dimension-agnostic versions of MetaBO in the paper. Nevertheless, we agree that this is an interesting route of research which we consider to address in more detail in future work.
>  - We now plot the results in log-scale.
>
> 3.) Applicability of MetaBO regarding test- and training-time
> -------------------------------------------------------------
> We would like to point out that NAFs can be used as a plug-in feature in any BO framework as it has the exact same interface as standard AFs. Furthermore, gradients for AF-optimization can be obtained effortlessly using automatic differentiation frameworks (we used the standard PyTorch framework for our implementation). To demonstrate that test-time runtime is not increased considerably compared to standard EI and to show that MetaBO scales much better w.r.t. the amount of source data compared to TAF, we added a table (App. A.3, Tab. 3) to our paper which compares the presented AFs with respect to test-time runtime. Regarding training time, we would like to point to the second paragraph of Sec. 5, where we now detail the computational resources for NAF-training.

---

### Official Review · AnonReviewer1 · 2019-10-24
**Official Blind Review #1**

**Rating:** 8

**Review:**


This paper proposes a framework for meta learning neural acquisition functions for the Bayesian optimization of various underivable functions. The neural acquisition functions are learned using proximal policy optimization in an outer loop on different problems on the same domain, and the learned acquisition function can be deployed at test time in a practically vanilla Bayesian optimization procedure. The authors demonstrate the performance of the method through benchmarks on four problems.

I recommend that this paper be accepted for publication. The paper is well written and it proposes a novel direction for research. However, I think that the authors should look further inside their newly designed acquisition functions, not merely treat them as black boxes. Find below some questions and comments.


Due to the inclusion of the sample position x in the state tuple, I am curious as to what the authors think is the difference between their method and a learning-to-learn type of approach. Is the acquisition function learning to favor specific zones in the search space based on previous experiments? Some more experiments or insights on this would be useful to better understand what makes this method succesful.

Why was a categorical distribution used for the policy? These samples are located in D, aren't you getting rid of information by assuming they are completely independent? Aren't you also biasing the distribution by adding the local maxima to the set of ξ (Xi)?

Also, the right-most block in Figure 1 shows a continuous probability distribution, which is incorrect. If the distribution is indeed categorical, there is no continuity between points.


Minor mistakes:

- page 5, paragraph 3: "This choice does not penalize explorative evaluations which do not yield and immediate improvement" should read "an immediate improvement"
- Figure 4b, MetaBO-50 is missing

***********
Post rebuttal:
************

I have read the other reviews and the various replies by the authors. I'd say you did a good job in answering most questions and added a lot of valuable information in the appendices. I maintain my score.

**Experience Assessment:**

I have published one or two papers in this area.

**Review Assessment: Checking Correctness Of Derivations And Theory:**

N/A

**Review Assessment: Checking Correctness Of Experiments:**

I carefully checked the experiments.

**Review Assessment: Thoroughness In Paper Reading:**

N/A

---

> ### Author Response · Authors · 2019-11-14
> **Author response to official blind review #1**
>
> Thanks for your very positive feedback and the acceptance score. We gladly answer the remaining open questions. We highlighted the updated sections in yellow in the updated PDF.
>
> 1.) Gain insights into behavior of NAFs
> ---------------------------------------
> We provide new experiments to give more insights into the behavior of our neural acquisition functions (NAFs). The results show that MetaBO's NAFs indeed learn representations that go beyond standard AFs combined with a prior over x. Please refer to Appendix A.1 of the updated PDF for details.
>  - We devised two one-dimensional toy problems (Rhino-1 (App. A.1, Fig. 5), Rhino-2 (App. A.1, Fig. 6)) to demonstrate that MetaBO learns to use non-greedy evaluations in the beginning of an episode to obtain high information gain (rather than low regret) about the target function. This results in more efficient search strategies compared to approaches which simply favor specific zones in the search space.
>  - Note that this effect can already be observed in the original results in our paper (Fig. 2, Fig. 3(a)), where MetaBO starts episodes with evaluations yielding higher regret than other pre-informed AFs (TAF) but quickly surpasses their performance by using the information obtained through these non-greedy evaluations.
>  - We further implemented two additional easily-interpretable baseline methods (GMM-UCB, eps-greedy) as proposed by AnonReviewer3 which rely solely on a prior over x. The results (App. A.1, Fig. 7) show that such simple approaches are not able to reach MetaBO's performance, underlining that MetaBO produces more sophisticated search strategies.
>
> 2.) Difference to learning-to-learn
> -----------------------------------
> Regarding your question of the difference of our approach to a learning-to-learn type approach such as Chen et al. [1], we would like to point you to our answer to AnonReviewer2, where we discuss this question in detail.
>
> 3.) Architecture
> ----------------
> We thank you for the remark that the right-most panel of Fig. 1 is inaccurate. It indeed shows a continuous distribution, while our policy defines a categorical distribution. However, our NAFs can be evaluated at any point in the domain and our method does indeed use an adaptive grid \xi_t to form this categorical distribution from the AF outputs during training. We wanted to emphasize this through the shaded area in our figure. We improved Fig. 1 in the PDF to remove the inaccuracies and extended and improved our description of our architecture in Sec. 4. We hope that the additional explanations help to clarify your questions.
>
> 4.) Minor mistakes
> ------------------
> We corrected the minor mistake. Furthermore, we explained that we did not carry out expensive hardware experiments for MetaBO-50 because it did not show promising performance in simulation compared to the full version of MetaBO.
>
> References:
> [1] Chen et al., "Learning to learn without gradient descent by gradient descent", ICML 2017

---

### Author Response · Authors · 2019-11-15
**Code update, Minor corrections**

Revision 15th Nov, 2019.
- We updated the code in the anonymous repository.
- We corrected some minor spelling mistakes in the first revision version of the PDF.

---

### Decision · Program_Chairs · 2019-12-19

**Decision:**

Accept (Spotlight)

**Comment:**

This paper explores the idea of using meta-learning for acquisition functions. It is an interesting and novel research direction with promising results.

The paper could be strengthened by adding more insights about the new acquisition function and performing more comparisons e.g. to Chen et al. 2017. But in any case, the current form of the paper should already be of high interest to the community